# Outcome measures for young people with adolescent idiopathic scoliosis: A qualitative exploration of healthcare professionals' perceptions and practices

Samia Alamrani[1,2]*, Adrian Gardner[3], Deborah Falla[1], Emily Russell[4], Alison B. Rushton[5], Nicola R. Heneghan[1]

1 Centre of Precision Rehabilitation for Spinal Pain (CPR Spine), School of Sport, Exercise and Rehabilitation Sciences, College of Life and Environmental Sciences, University of Birmingham, Birmingham, United Kingdom, 2 Physical Therapy Department, College of Applied Medical Science, University of Tabuk, Tabuk, Saudi Arabia, 3 Spine Unit, The Royal Orthopaedic Hospital NHS Foundation Trust, Northfield, Birmingham, United Kingdom, 4 Oxford University Hospitals NHS Foundation Trust, Oxford, United Kingdom, 5 School of Physical Therapy, Western University, London, Ontario, Canada

* sxa1258@student.bham.ac.uk, salamrani@ut.edu.sa

**Data Availability Statement:** All relevant data are within the manuscript and its Supporting Information files.

## Abstract

### Background

Limited knowledge exists on current use of patient reported outcome measures (PROMs) and performance measures for adolescents with idiopathic scoliosis (AIS), as well as health care professionals' (HCPs) perceived barriers and facilitators towards their use. This study's objectives were: 1) to explore current practice of HCPs when assessing outcomes for AIS 2) to understand perceived barriers and facilitators of HCPs to use PROMs 3) to understand perceived barriers and facilitators of HCPs to use performance measures.

### Methods

A qualitative study recruited a purposive sample of HCPs from a tertiary hospital in the United Kingdom. Mean years of experience managing individuals with AIS was 11.8 years; and included surgeons, physiotherapists and nurses, educated at Bachelor, Masters and Doctoral level. Consent to participate and demographic information were collected in advance of the interviews. In-depth, virtual semi-structured interviews were informed by a topic guide based on current evidence. Interviews of approximately 45 minutes were audio and video recorded and transcribed verbatim alongside written field notes. Data were coded and analysed using inductive thematic analysis, involving researchers with topic and methodological expertise and input from a patient representative.

### Results

Two themes emerged regarding current practice of using PROMs routine practice and personal evaluations. Four themes emerged as barriers to using PROMs for individuals with AIS: priority and support (e.g., HCPs focus on providing care), practical challenges (e.g.,

**Funding:** The author(s) received no specific funding for this work.

inadequate PROMs), patient-related challenges (e.g., patient preferences) and knowledge, education, and perceived value. Two themes emerged as facilitators: quality existing measure (e.g., sufficient psychometric properties), and priority and support (e.g., research department/culture). Themes for barriers to use performance measures were practicality (e.g., need physical space) and perceived value and knowledge (e.g., PROMs are more important), while the one theme for facilitators was practical consideration (e.g., acceptability).

## Conclusions

Although HCPs perceived the value of using outcome measures, current practice indicates limited use for individuals with AIS. The findings revealed different barriers and facilitators to implement PROMs in practice. Adopting performance measure are limited due to lack of knowledge and perceived value alongside the practicality, while considering practical factors can improve the use of these measures in practice.

## Background

Adolescent idiopathic scoliosis (AIS) is the most common spinal deformity among the paediatric population, with an overall prevalence ranging between 1 to 3% [1, 2]. Nearly 10% of individuals with AIS require a form of intervention that may eventually lead to spinal fusion surgery [3]. Outcome assessment is essential in routine clinical practice, as it facilities the decision-making process, improves communication between individuals with AIS and clinicians, evaluates effectiveness of treatment, compares different treatments and monitors progress over time [4, 5].

Paediatric research has shown that children are able to provide reliable information about themselves, and that their point of view is more important than that of their proxy adults [6]. Also, adolescence is a discrete age group that have distinct characteristics, with unique emotional and developmental stages [7]. Therefore, assessment of their outcomes should relate to their abilities, interests, and social relationships [7].

Although there is increased interest in the use of outcome measures (OMs), current knowledge is scarce about current practice in the assessment of outcomes for individuals with AIS. The assessment of outcomes in AIS has conventionally focused on the amount of correction of spinal curve radiographically [8], with little attention to other outcomes. A recent systematic review identified that the majority of studies investigating measurement properties of OMs in AIS, are for patient reported outcome measures (PROMs), with few studies evaluating performance measures [9] that therefore limits their use for individuals with AIS.

Although PROMs are useful tools for gathering information about a patient's health status and quality of life from their own perspective, their responses may be influenced by their perception of change as well as other variables such as pain and psychological stress, which are frequently reported in this population [9]. Individuals with AIS often experience pain prior to and following surgery that contribute to avoidance of certain activities [10]. Therefore, a PROM's assessment of the functional performance of an individual may not capture the complete picture of an individual's ability. A clinical measure, such as radiographs and muscle strength, give an indication about impairment of structure or function, but does not fully capture the functional limitations of an individual [11]. On the other hand, performance measures

such as walking speed or endurance testing afford a more comprehensive assessment of function [11]. Performance outcome measures can be used to evaluate the impact of interventions such as surgery and exercise on specific functional domains such as mobility, strength, endurance, and balance, which provide a holistic understanding of patient's condition [12]. Nevertheless, variables that could improve or limit the usefulness of such measurements in individuals with AIS remain unknown.

Various barriers to the use of OMs have been identified in other fields, such as clinician self-judgment, time constraints, lack of knowledge and lack of resources [13]. Facilitators have also been identified, including knowledge and training to use an OMs, availability of the OMs, and guidelines recommending its use [14–17]. The existing research predominantly emphasizes PROMs in individuals with AIS but lacks substantial exploration into the utilization of performance measures and the impact of demographic factors, such as age, and curve severity, on the application of OMs in AIS. To date, no research has explored current practice of using OMs for individuals with AIS, or health care professionals' (HCPs) perceived barriers and facilitators of all forms of OMs.

## Objectives

1. To explore the current practice of the HCPs when assessing outcomes for individuals with AIS

2. To understand the perceived barriers and facilitators of HCPs to use PROMs.

3. To understand the perceived barriers and facilitators of HCPs to use performance measures.

## Methods

### Ethics approval

Ethical approval to conduct the study was sought from the Health Research Authority and Health and Care Research Wales approval (REC reference: 21/WM/0076). All study participants provided their written informed consent to participate before conducting the study.

### Study design

This qualitative study is a component of a larger study that evaluated the content validity of the Scoliosis Research Society 22 revised (SRS-22r), from the perspective of both patients and HCPs [18], reported in line with the consolidated criteria for reporting qualitative studies (COREQ) [19]. The descriptive phenomenology was chosen as methodology for this study, as it focuses on describing the universal essence of an experience as lived by individual [20]. In this study, a pragmatic world view was adopted, emphasizing the significance of choosing research methods based on their suitability for addressing specific research questions rather than strictly adhering to philosophical debates [21].

### Settings

Due to COVID-19 pandemic restrictions that were in place at the time of data collection, the in-depth semi-structured interviews with HCPs were conducted virtually via Zoom. Zoom is considered as a highly satisfactory software alternative to face-to-face interviews, because of its ease of use, cost-effectiveness, and security options [22]. The timing of interviews was selected based on participants preferences and interviews lasted a mean of 45 minutes (SD = 9.7).

## Participant selection

Participants were invited to participate in the study via email. They were recruited from a tertiary scoliosis centre in the United Kingdom. Interviews were conducted between July and September 2022. A purposive sample of HCPs, including physiotherapists, nurses, and surgeons, who were part of health care team for individuals with AIS was recruited [23]. Participants were considered eligible if they had experience in managing AIS.

## Data collection

The interview topic guide (S1 Appendix) was developed by the research team using current evidence [21]. Field notes were assimilated following the interview. Participants were all unknown to the interviewer prior to the interview. The interviews were audio and video recorded, then transcribed verbatim through official transcription services. Transcripts were emailed to participants to allow corrections to be made or to add any further details. Participants were given two weeks to make any alterations or suggest changes. Demographic information of the participants, along with the consent forms were collected electronically prior to commencement of interviews.

## Data storage and management

In compliance with the Data Protection Act 2018, all study investigators adhered to guidelines governing the collection, storage, processing, and disclosure of personal information. Personal data underwent coding and depersonalization, replaced by participant identification numbers, and were securely stored on a password-protected computer at the University of Birmingham. The data's secure maintenance included encrypted digital files, password-protected folders, and storage media, with access limited to the research team. The linking code was kept separately in a secure, encrypted location. The data will be retained for 10 years in alignment with the University of Birmingham's Research Governance procedures, accessible solely to the research team.

## Data analysis

The lead researcher (SA) listened to the transcripts and performed the initial coding, which was reviewed and discussed with qualitative experienced co-authors, to enhance the credibility of the study findings. To ensure that participants' experiences and perspectives are accurately reflected, allocation of codes and theme alignment with participants quotes were evaluated. A saturation table was created to ensure that concept saturation was achieved (Table 1), defined as the point where two consecutive transcripts failed to elicit any new themes [24].

Inductive thematic analysis following the Braun and Clarke framework [25], was used to analyse the data (Fig 1).

## Research team and reflexivity

An experienced musculoskeletal physiotherapist researcher (SA) conducted the interviews, with support from the co-authors who have expertise in conducting spinal and qualitative research. Further, a spinal surgeon with experience working with AIS was involved along with a patient and public involvement (PPI) representative. No relationship was established with participants prior to the commencement of the interviews. Participants were informed about the professional background of the interviewer and that the study was part of a PhD thesis. All authors actively participated in the analysis of data and derived themes and subthemes.

**Table 1. Saturation table showing that theme saturation was achieved.**

| Themes | Subthemes | Interview number | | | | | | |
|---|---|---|---|---|---|---|---|---|
| **Objective 1: Current practice** | | **1** | **2** | **3** | **4** | **5** | **6** | **7** |
| Routine practice | Standard practice in hospital | ✓ | | | | | | |
| | To monitor outcomes | ✓ | | | | | | |
| Personal evaluation | Personal questions | ✓ | | | | | | |
| | Use patients goals and needs | | ✓ | | | | | |
| | Rating of outcomes | | ✓ | | | | | |
| **Objective 2: Barriers and facilitators to use PROMs** | | | | | | | | |
| **Barriers** | | | | | | | | |
| Priority and support | Difficult to implement | | ✓ | | | | | |
| | Focus on providing care | | ✓ | | | | | |
| | Not informing practice | ✓ | | | ✓ | | | |
| Practical challenges | No time | | | | | | | |
| | Inadequate PROMs | | ✓ | | | | | |
| | Logistic provisions | | | ✓ | | | | |
| Patient-related challenges | Potential invalidity | | | | ✓ | | | |
| | Patient preferences | ✓ | | | | | | |
| | No discharge assessment | | ✓ | | | | | |
| Knowledge, education and perceived value | Lack of knowledge | | | ✓ | | | | |
| | Questioning importance | | | ✓ | | | | |
| | Need of education | | | ✓ | | | | |
| **Facilitators** | | | | | | | | |
| Quality exiting measure | Sufficient psychometric properties | | | ✓ | | | | |
| | Updated and modernised | ✓ | | | | | | |
| | Personalised and relevant | ✓ | | | | | | |
| | Comparable and widely acceptable | | | | | ✓ | | |
| | Simple and easy | | | ✓ | | | | |
| Priority and support | Research department | ✓ | | | | | | |
| | Scoring system | | | ✓ | | | | |
| | Show difference in practice | | | ✓ | | | | |
| **Objective 3: Barriers and facilitators to use Performance measure** | | | | | | | | |
| **Barriers** | | | | | | | | |
| Perceived value and knowledge | Patient reported measure more important | | ✓ | | | | | |
| | Performance indicators differ across individuals | | | | ✓ | | | |
| | Unaware about use performance measure | | | | ✓ | | | |
| Practicality | Need physical space and time | | | ✓ | | | | |
| **Facilitators** | | | | | | | | |
| Practical consideration | Relevant | | | | ✓ | | | |
| | Simplicity and acceptability | | | ✓ | | | | |
| **No. of new codes appearing in each interview** | | 8 | 7 | 10 | 5 | 1 | 0 | 0 |
| **% Of total new codes (Total = 34)** | | 23 | 20.5 | 29.4 | 14.7 | 2.9 | 0 | 0 |

## Trustworthiness

Many strategies have been employed to enhance trustworthiness of the study results [26]. Credibility of the study was determined by using prolonged engagement with participants, promoting a deeper understanding of their experiences [26]. Member checking was also employed to validate findings [26], by asking participants to review and confirm the accuracy of the study findings The research team, consisting of researchers with both topic and

| Step 1 | • Familiraisiation with data by reading and re-reading it several times |
| Step 2 | • Generation of initial codes |
| Step 3 | • Searching for themes |
| Step 4 | • Reviewing themes to ensure they accurately reflect the data |
| Step 5 | • Defining, and naming themes |
| Step 6 | • Production of the report |

**Fig 1. Thematic analysis steps.**

methodological expertise, participated in data analysis enhancing the robustness of the interpretations [26]. The transferability of the study findings was ensured by presenting findings with a focus on providing a "thick description" of the context and participants' experiences [26].

## Results

### Participants

Of the 13 participants invited, seven agreed to participate in this study. Participants included four females and participants had a mean age of 42 years (SD = 7.9). Professions of participants included surgeons (n = 3), nurses (n = 2) and physiotherapists (n = 2) with a mean of 11.8 years (SD = 4.9) of experience working with individuals with AIS. Their highest qualification was a doctorate (n = 1), followed by a master's degree (n = 4) and bachelor's degree (n = 2). Different themes and subthemes were identified from participants for each objective of the study (S2 Appendix).

### Objective 1: Current practice

Two themes related to current practice of using OMs were identified from participants' descriptions: routine practice and personal evaluation with associated subthemes.

## 1. Routine practice

**Standard practice in hospital.** Participants described that the current use of OMs is standard routine practice within the hospital, as per the national guidance of the British Spine Registry. The PROMs used for individuals with AIS is the Scoliosis Research Society-22 questionnaire. Individuals with AIS routinely complete the questionnaire prior to the appointment with the HCPs. It also sent to the individual/parent email address automatically at specific time points to be completed remotely, to monitor changes over time.

> "*Actually, we use them for all the patients that come through as the pre-operative assessment. But it's only for research purposes. I don't use that in the clinical setting. I don't look at the score and decide on whether they have an operation or not. It is only for research purposes*" (P6- surgeon)

**To monitor individuals with AIS outcomes.** Participants perceived a strong benefit from using PROMs in this population to help them identify patient problems, monitor long-term outcomes as well as giving a good scope and understanding of patient's feelings.

> "*If a patient, for example, is being monitored for a few years, then prior to proceeding to need surgery, we could have maybe three years of data and seeing how the results have deteriorated or maybe not deteriorated, but we would see more of.. able to see a larger pattern*" (P1-Nurse).

## 2. Personal evaluation

Whilst none of the participants reported use of any PROMs, they reported use a range of personal assessment tools to evaluate patients' outcomes.

**Personal questions.** Participants reported using personal questions to assess patients' outcomes. At every visit to the clinic, patients were asked about their life with scoliosis, how they feel about it, and how much activity they can perform.

> "*Only through questions during the history, not through a questionnaire. So, we do evaluate that, but not through a questionnaire that gives a numerical score. We would ask questions and have a more qualitative answer to that*" (P4-Surgeon)

**Use patient goal and need.** Physiotherapists reported using patient goals and needs as outcome measures to evaluate how their outcomes are changing.

> "*Just with specific goals, really. So, we'd use patient-specific goals to identify what they want to achieve, what they're not able to do, and then use that just as a specific measure: measure as to how much they're improving*". (P5-Physiotherapist)

**Rating of outcomes.** Physiotherapists reported asking patients to rate their level of pain and function, as well as to rate the percentage of improvement with regards to the return to normal function or activity.

> "*We're not using any really.. . . But really I'm probably getting them to rate their pain scale on like a one to 10 maybe. I'm also getting them, . . . to rate how much better they're feeling. . .So out of 100%*". (P2-Physiotherapist)

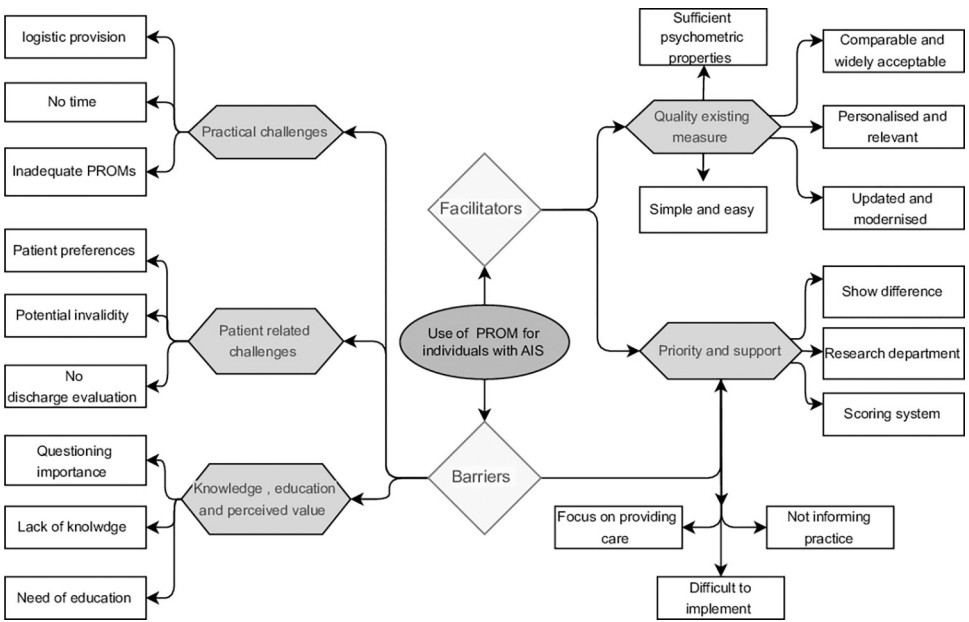

**Fig 2. Barriers and facilitators identified from participants data regarding use of patient reported outcome measure for individuals with AIS.**

## Objective 2: PROM's barriers and facilitators

Four themes emerged from interviews as barriers to use PROMs for individuals with AIS while two themes emerged as facilitators, with multiple subthemes also identified. (Fig 2) represents the themes and subthemes.

### Barriers

**1. Priority and support.** The perceived importance of using PROMs in practice, and the lack of support was recognised as a barrier for HCPs in adopting the PROMs in practice.

*Focus on providing care.* Participants prioritised providing care to their patients rather than evaluating their outcomes using standardised measures.

> "*I think you're kind of focused on providing physio*" (P2-Physiotherapist)

> "*Of course, as a clinician, although it's nice to be involved in research, our primary focus is patient care*" (P4-Surgeon).

*Not informing practice.* Participants perceived that the use of PROMs is not informing practice for AIS compared with clinical measures, which are determinant of the patient's future. Participants stated that the use of PROMs is for research purposes only and scores obtained did not inform their practice or clinical decision making. Further, they think that a PROMs that evaluates quality of life is more valuable for those with AIS than radiological measures.

> "*They need to take formal measurements of your growth plates and your degrees of curvature. . . .to decide whether it's going to be conservatively managed or surgically managed. That really dictates their future.*" (P2-Physiotherapist)

*Difficult to implement*. Participants perceived that the use of PROMs is difficult to be used in practice owing that to the lack of support, such as research team assigned to collect PROMs from patient. This in turn impacts compliance.

"*Outcome measures have come in and we have certainly tried to use them in certain things. . .. But it is very difficult to implement that, people's compliance with it, get them trained up on it*" (P2-Physiotherapist)

**2. Practical challenges.** Participants discussed a variety of practical challenges that impact the adoption of PROMs in routine clinic practice.

*No time*. Participants discussed that there is no time as the biggest challenge to use PROMs. Clinician-time as well as patient-time and busy clinics were reported.

"*I think sometimes as well it's from an environmental perspective and a perspective of just time restraints, patient-wise and clinician-wise. . . "I'm sorry, I can't fill that in today because I've got to go, because I've got another healthcare appointment." I think it's both ways*" (P1-Nurse)

*Inadequate PROMs*. Participants perceived that the qualities of the currently used PROMs are inappropriate affecting use in practice. They reported that some PROMs are lengthy, its language is outdated, and irrelevant to the current young generation, which in turn affect patient's ability to fill questionnaire.

"*I mean it is quite a lengthy questionnaire. It is a number of pages and they, especially for our young adults, the AIS patients*" (P7-Nurse)

"*Actually, the language of the SRS study is quite difficult to interpret if you're a younger adolescent. . . I think some of the language in it is quite outdated in comparison to language that we may use now*" (P1-Nurse)

*Logistic provisions*. This was reported by HCPs as barriers that enable the use of digital versions of PROMs. Further, the system that is used to collect data from patients is "not user-friendly" which limit usefulness of the data.

"*The Amplitude system is not the most user friendly to get at and I don't bother to look at it because I'm too busy*". (P3-Surgeon) "*I think IT and logistical provisions are the constraints*". (P6-Surgeon)

**3. Patient-related challenges.** This theme described the challenges raised by HCPs to use of PROMs regarding patient's perspective.

*Potential invalidity*. This refers to the concern of HCPs regarding a patient's understanding of SRS-22r questions. They reported that individuals with AIS may ask for help or clarification from their parent or care team to complete the questionnaire, which affects the validity of the collected data.

"*You're having a parent talk it through and then also if you're talking it through, then you naturally may bring in your own judgement or suggestion of what something could be, which again, takes away the validity of it because one understanding, how they interpret it*". (P1-Nurse)

*Patient preferences*. Participants discussed that some patients don't want to "talk about their health condition" or "participate" in research studies which impact the use of PROMs.

"*Some of them don't want to talk about their condition. Some of them absolutely. . . their parents don't want them to be engaged with the research as well*" (P7-Nurse)

*No discharge evaluation*. Physiotherapists highlighted that the last treatment session on many occasions is unknown. As patients or their parents may cancel or stop the treatment without prior knowledge of the physiotherapist. Therefore, no discharge assessment can be performed. This will limit physiotherapist's ability to provide PROMs to patients to evaluate their outcomes.

"*So, there's not like this formal today's the end of your treatment so then I'm going to provide you with all these questionnaires. It's a bit more fluid and that's probably our biggest limitation*" (P2-Physiotherapist)

**4. Knowledge, education, and perceived value.**   This theme described the barriers recognised by HCPs with regards to the knowledge, education and the value of using PROMs in routine practice.

*Questioning importance*. Participants questioned the importance of using PROMs in their clinical practice.

"*I think if we were to implement some sort of outcome measure, generally, there's quite a lot of effort behind why it's important and how it will improve my life, how it will improve my patient's lives*" (P3-Surgeon)

*Lack of knowledge*. Participants reported that they did not know about the PROMs questions, or what is the PROMs score could tell.

"*Again, I don't really understand the questionnaire. Nobody has ever explained it to me, and I've been in the role for eight years. You get given a percentage at the end of it and I have no idea what that percentage means*" (P7-Nurse)

*Need of education*. Others recognised that that they have "never reviewed their practice" with individuals with AIS. They also reported that HCPs "don't want to change" and that a "lot of education" is required to implement changes in practice.

"*My experience of consultants, people like me and colleagues, is that you do it one way and you don't ever change. . .someone like me to change what they do to a different thing, can take quite a lot of education*" (P3-Surgeon)

## Facilitators

Two main themes were identified as facilitators to use PROMs for individuals with AIS: quality existing measures and prioritization and support.

**1. Quality existing measure.**   This theme includes the characteristics identified by HCPs that qualifies that the PROMs is a "good existing measure" to enable its use for individuals with AIS.

*Sufficient psychometric properties.* Participants reported that the PROMs should have certain characteristics to facilitate its use in practice. Such as "being repeatable", "sensitive to change", and having "minimal number of errors".

"*I think the ideal thing would be something that means something to the patient, that changes with intervention or disease progression. So, you can show there's a change. That's easily understandable. That is easily measurable, however you measure it. It's repeatable, has minimal amount of error in it*". (P3-Surgeon)

*Comparable, and widely acceptable.* HCPs reported that the PROMs should be widely accepted to enable results to be compared between different treatments and among patients in different countries.

"*If you found a questionnaire that had good psychometric properties that no-one else used, then, if you were to do a study on it, it would be very difficult to compare your results to other studies*" (P4-Surgeon)

*Personalised and relevant.* These features of the PROMs were highlighted by HCPs and it was stated that it should be personalised to individuals with AIS.

"*I think if we're going to measure outcome, that has to be front and centre... But it also then has to be personalised to that individual.*" (P3-Surgeon)

*Updated and modernised.* Participants highlighted that they need a PROMs that is updated to the young generation to facilitate its use and have better engagement.

"*I think if we had a different type of questionnaire that was more able to engage the appropriate population and updated, easy to understand terminology... we would get more accurate data... there's so much changes within practice ... and the mindsets of our young people as well, that now is the time for it to kind of be modernised*" (P1-Nurse)

*Simple and easy.* Participants reported they need a questionnaire that has simple and easy wording for their young individuals with AIS.

"*With the younger adolescents, you just need to keep it fairly simple wording*".
(P5-Physiotherapist)

"*I want something that is easy to use on my patients, easy to interpret for me. Something that is readily available*". (P7-Nurse)

**2. Priority and support.** This theme described what HCPs need to support and prioritize the use of PROMs in routine clinical practice.

*Research department.* Participants reported the need of having a big research department to help improve the application of OM in practice, by employing staff designated to collect data from patients without interfering with patient or clinician time.

"*Those are all things that we wish we could do, but we need a bigger research department to be able to achieve this*" (P1-Nurse)

*Scoring system.* Participants discussed the need of having a scoring system that could help in evaluating treatment effectiveness and identify patient problems.

"*.. it's good if we have a scoring system, if you were to compare either two treatment modalities or we want to look at the outcome of a particular..That's important. . .the score domains are useful to identify any particular problem with the patient..*". (P6-Surgeon)

*Show difference in practice.* Participants reported that to enable use of PROMs in practice, that OMs should show a difference in their routine clinical practice.

"*I think if we had an outcome measure that could show there's a benefit to the surgery, . . . then go to my colleagues and say, this is important, because look, this shows what you do makes a difference. That would be worth having. . .*"(P3-Surgeon)

## Objective 3: Barriers and facilitators to use performance measure

### Barriers

Two themes emerged as barriers and one theme was identified as a facilitator regarding use of performance measure in routine clinic practice among in individuals with AIS. (Fig 3) shows the identified themes and subthemes.

**1. Perceived value and knowledge.** *PROMs are more important.* Participants perceived that the performance measure is not important as a self-reported measure.

"*There are some situations where performance based is important to take decisions. But in scoliosis it's a little more subjective in terms of outcomes. If a 14-year-old comes back and says, "I'm back playing football," at the six-month mark, that to me is more important than what the six-minute walking score is*" (P6-Surgeon)

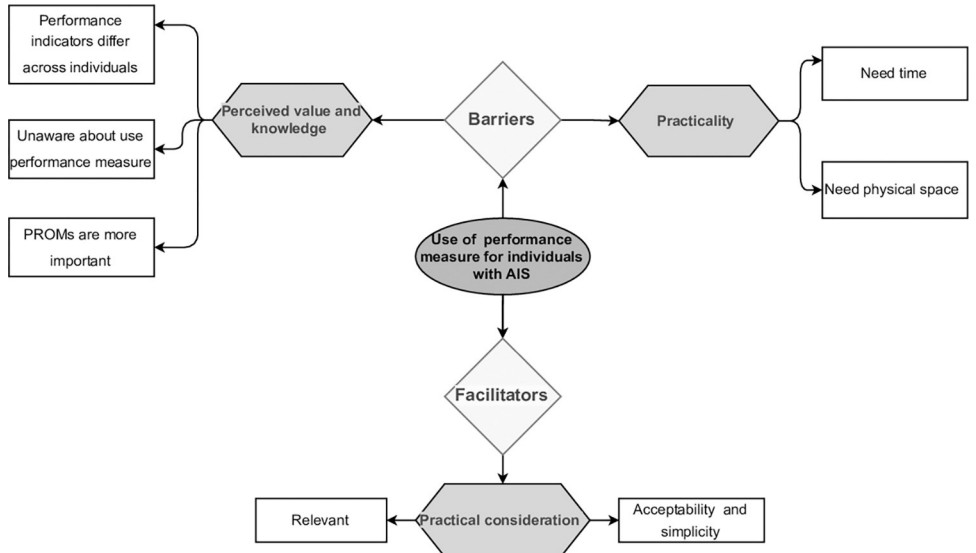

**Fig 3. Barriers and facilitators identified from participants data regarding use of performance measures for individuals with AIS.**

*Performance indicators differ across individuals*. Participants discussed that although the performance measures are applicable to individuals with AIS, the indictors of performance measures differed across individuals.

"*The issue there would be what performance measure do you put most emphasis on, in an individual patient? There are so many different performance indicators that you could use, that one could potentially use. . ...the issue, is what's important to one child will be very different to another child*" *(P4-Surgeon)*

*Unaware about use performance measure*. Study participants discussed that their knowledge and experience about using performance measures in those with AIS is limited.

"*They do this quite a lot don't they, with shoulders, hips and knees and other elements, don't they? But I have to be honest, I haven't seen it routinely used within adolescent spinal deformity*" (P1- Nurse)

**2. Practicality.**    *Physical space and time*. Participants reported that the main barriers to use performance measures in clinics for individuals with AIS, are the physical space and the time needed to perform it.

"*I think if we're thinking about some sort of physical activity, then the problem we'd have with AIS, is physical space. I can do a paper form in clinic because I give it to the kid who is sitting on the chair. But if I want someone to, I don't know, walk them down the corridor 20 times.. it's the practicalities of how you do that and how I do it with 30 children at the same time essentially. . ..That would be the thing for me*" (P3-Surgeon)

## Facilitators

**1. Practical consideration.**    This theme related to the characteristics of performance measures that may enhance its use in clinical practice per participants reports.

*Relevant*. Participants reported that the performance measure should be relevant to the individuals with AIS.

"*I would group the patient. If you're looking at scoliosis, I would look at adolescent idiopathic, versus neuromuscular, versus syndromic. Then I might choose an outcome measure that would be relevant to that particular (sub-) group of patients*" (P4-Surgeon)

*Simplicity and acceptability*. The simplicity and acceptability of performance measures was also discussed by participants, that it is acceptable to individuals with AIS and can be applied quickly in the clinic were facilitators to use performance measures.

"*If we were just looking at, I don't know, say five key areas and what's their flexibility, what are they able to achieve in regard to, . . . very simplistic system, then that could be applied fairly quickly*" (P1-Nurse).

## Discussion

This is the first qualitative study that has evaluated the perspectives of HCPs in the use of OMs among individuals with AIS. Participants in this study were derived from different professions,

all with considerable experience of AIS. Despite the perceived importance and value of using OMs, findings indicate limited use of OMs in routine clinical practice. Themes identified as barriers to the use of PROMs are knowledge, education and perceived value, priority and support, patient related, and practical challenges. Having a quality existing measure and support were identified as the main facilitators to use PROMs for individuals with AIS. Barriers to adopting performance measures for individuals with AIS are perceived value and knowledge as well as practicality. Facilitator that promote use of performance measures for individual with AIS were being relevant, simple and acceptable to individual with AIS.

## Current practice

The British Spine Registry (BSR) was established in 2012 in UK, to monitor the outcome of patients undergoing spinal procedures, including AIS. Despite the inclusion of SRS-22 and Visual Analogue Scale, our findings suggest that the system is not fit for purpose in the AIS population. Participants reported that the system is not user-friendly, they lack understanding of the included questionnaires and what scores mean. According to a recent study, the BSR response rate dropped from 58% completed preoperatively to 25% one year afterwards [27]. Furthermore, a low response rate to automated BSR email reminders was also observed, with 56.7% at 6 weeks dropping to 48% at a 1 year follow up [28]. As per our participants' reports, a variety of factors, including a lack of time and a burdensome questionnaire, were recognised as obstacles to compliance [27].

In this study, participants valued assessment of pain and function for those with AIS. They recognised the importance of using PROMs in practice to monitor individuals with AIS outcomes. However, all participants reported using informal self-reported measures, and they link it to patients' goals and needs, a finding consistent with other research in different patient populations [14, 16, 17].

## Barriers and facilitators to the use of PROMs

Inadequate PROMs design and content was reported as a practical challenge and a barrier to use of PROMs in practice. The widely used SRS-22 was described by participants as being irrelevant, lengthy, and its language outdated for young individuals such as those with AIS. They were also concerned about validity of findings where patients needed assistance with understanding and completion. Existing evidence suggests that there are no differences in SRS-22 scores if completed by the individuals with AIS or their parents [29]. Meanwhile, a study has been designed to evaluate content validity of the SRS-22r for those with AIS [18]. Content validity is the first and most important measurement property that should be assessed before a PROMs can be used in practice [30].

Other reported barriers included non-participation in research with families not wishing to talk about their condition, restricting use of PROMs in practice. This is consistent with reports from previous studies among individuals with AIS waiting for surgery, as those with AIS tend to hide their condition from others [10, 31]. For this current study, participants reported focusing on providing care to their patients, and the use of PROMs is difficult in daily practice, consistent with other reports [17]. They perceived that the use of PROMs is for research purposes only, and that they need to see change in practice. consistent with the national guidance that recommended that HCPs should understand how the use of PROMs can be beneficial and lead to changes in their practice [32]. Participants, on the other hand, indicated that they require a large research department dedicated to collecting data from patients in order to improve its usage in practice. The wider literature indicated that 55% of HCPs require support from staff in data collection [33]. Lack of knowledge or the perceived value of a PROMs was

also reported as a barrier [4, 5, 33, 34], however the provision of training, education and support on the value of using PROMs and how to use them were considered facilitators [5].

A quality existing PROMs is a key theme that has been recognised as facilitator to PROMs use in practice. Sufficient measurement properties such as validity and reliability were discussed and are widely recognised in the literature [35]. Other reported PROMs characteristics included being relevant, simple, easy to use and comparable with other reports [36]. This indicates that HCPs value the use of PROMs, however they perceived that the available PROM lacks the characteristics of a quality measure. Therefore, it has been recommended that healthcare organisations should invest time and resources, in designing PROMs to facilitate use in practice [5].

## Barriers and facilitators to the use of performance measures

Participants reported minimal knowledge and experience about performance measures, placing greater value on PROMs in AIS. Practical issues of such measures were identified as barriers i.e., the time and physical space needed; consistent with PROM studies where time to collect data acts as barrier [15, 16]. These findings are consistent with previous reviews, where most of measurement properties studies identified were from a PROMs compared to performance measure [9]. However, it is well recognised that although PROMs are important, they measure only one aspect of a condition and it should be associated with physical performance assessment [37]. Performance based measures are commonly used in other patient groups, such as those with upper limb [38], knee and hip [39], and low back pain [12, 37, 40]. Up to now, little attention has been paid to the evaluation, applicability and acceptability of such measures to those with AIS; this would be a fruitful area for further work. Practical considerations are the main theme identified as facilitators to the use of performance measures for individuals with AIS.

Participants reported that to enhance its use, it should be acceptable, relevant, and applicable. Consistent factors such as feasibility and interpretability were identified as important to using performance tests in sport and exercise fields [41]. Ensuring that the measures are acceptable to the patient, as some patients and their families may not be willing to participate or results may not be accurate, which would make it difficult to draw valid conclusions from the data. Thus, patient and public involvement becomes an integral requirement in designing research studies and reporting results [42].

## Strengths and limitations

This study was designed and reported in line with published guidance of qualitative research (COREQ), and a published protocol [18]. All stages from design to interpretation involved experts in qualitative research, HCPs representation and a PPI representative, thus enhancing trustworthiness and credibility of findings. The outcomes of this study present the perspectives of experienced professionals with this specific group, reflecting various credentials and professions. The sample of this study was recruited from one tertiary hospital in UK, limiting transferability of the findings. This study was designed as focus group discussion [18], however, due to HCPs job commitments, and COVID restrictions in place, this was deemed impossible, and therefore online interviews were conducted [43].

## Implications for practice and future research

Using of OM for individuals with AIS has a variety of implications on clinical practice. It enables providing patient-centred care that may improve patient outcomes, and result in more effective use of healthcare resources [44]. Understanding of barriers and facilitators identified

in this study could inform the development of strategies and polices to promote the use of OMs for individuals with AIS. Future research could identify the long-term effects of using different types of OMs for individuals with AIS, on outcomes and quality of life of those individuals.

## Conclusion

Although HCPs perceived the value of using OMs, current practice indicates limited use among individuals with AIS. Knowledge, and support to use PROMs as well as practical and patients-related challenges work as barriers that constrain PROMs use, while providing support and having quality PROMs facilitate its use. Barriers to adopt performance measures are a lack of knowledge and perceived value alongside the practicality, while practical consideration as acceptability act as an enhancer to the use of performance measures among this population.

## Supporting information

**S1 Appendix. Topic guide.**
(PDF)

**S2 Appendix. Themes, subthemes and codes.**
(PDF)

## Author Contributions

**Conceptualization:** Samia Alamrani, Alison B. Rushton, Nicola R. Heneghan.

**Data curation:** Samia Alamrani, Deborah Falla, Alison B. Rushton, Nicola R. Heneghan.

**Formal analysis:** Samia Alamrani, Deborah Falla, Alison B. Rushton, Nicola R. Heneghan.

**Investigation:** Adrian Gardner, Alison B. Rushton, Nicola R. Heneghan.

**Methodology:** Samia Alamrani, Adrian Gardner, Deborah Falla, Emily Russell, Alison B. Rushton, Nicola R. Heneghan.

**Project administration:** Adrian Gardner, Deborah Falla, Alison B. Rushton, Nicola R. Heneghan.

**Supervision:** Adrian Gardner, Deborah Falla, Alison B. Rushton, Nicola R. Heneghan.

**Validation:** Deborah Falla, Emily Russell, Alison B. Rushton, Nicola R. Heneghan.

**Writing – original draft:** Samia Alamrani.

**Writing – review & editing:** Samia Alamrani, Adrian Gardner, Deborah Falla, Emily Russell, Alison B. Rushton, Nicola R. Heneghan.

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
