## [Decision Letter · Decision Letter 0]

7 Nov 2023

PONE-D-23-27580Outcome Measures for Young People with Adolescent Idiopathic Scoliosis: A Qualitative Exploration of Healthcare Professionals' Perceptions and PracticesPLOS ONE

Dear Dr. Alamrani,

Thank you for submitting your manuscript to PLOS ONE. After careful consideration, we feel that it has merit but does not fully meet PLOS ONE’s publication criteria as it currently stands. Therefore, we invite you to submit a revised version of the manuscript that addresses the points raised during the review process.

We look forward to receiving your revised manuscript.

Kind regards,

Abera Mersha, MSc.

Academic Editor

PLOS ONE

Journal Requirements:

Reviewers' comments:

Reviewer's Responses to Questions

**Comments to the Author**

1. Is the manuscript technically sound, and do the data support the conclusions?

Reviewer #1: Yes

Reviewer #2: Yes

2. Has the statistical analysis been performed appropriately and rigorously? 

Reviewer #1: Yes

Reviewer #2: N/A

3. Have the authors made all data underlying the findings in their manuscript fully available?

Reviewer #1: Yes

Reviewer #2: Yes

4. Is the manuscript presented in an intelligible fashion and written in standard English?

Reviewer #1: Yes

Reviewer #2: Yes

5. Review Comments to the Author

Reviewer #1: Overall comment:

Although the present study is having a very limited number of participants, this study has a significance for the HCPs that must be perceived as a valuable parameter for using Oms. The obtained results are of original research in this study and lay emphasis on the current practice of following and implementing HCPs. The study design and methodologies used herein are scientifically sound and are described in sufficient details. Furthermore, this study contributes to the aspect of encouraging all the participants (surgeons, nurses and physiotherapists etc.) to understand the importance of such data to overcome the barriers and improve the quality of life of individuals with AIS. Conclusions drawn upon by the authors conform in an appropriate fashion and are supported by relevant references. The ethics of experimentation and research integrity is satisfactorily applied in this study. The article adheres to appropriate reporting guidelines and community standards for data availability. Limited knowledge, and awareness to support the use PROMs by the healthcare individuals as well as the patients along with their facilitators, challenge such efforts and work as barriers that constrain PROMs use, while providing support and having quality PROMs facilitate its use to act as an enhancer to the use of performance measures among this population. Such works need to be carried out at a larger level and the present study can provide impetus to achieve this goal in near future.

Background:

The authors make two statements:

(lines 92-94) that “Nevertheless, variables that could improve or limit the usefulness of such measurements in individuals with AIS remain unknown.”

(lines 100-102) that “To date, no research has explored current practice of using OMs for individuals with AIS, or health care professionals’ (HCPs) perceived barriers and facilitators of all forms of OMs.”

These two statements stand diluted with their third statement:

(lines 98-100) that “Current research mostly focus on PROMs, with little exploration of the use of performance measures or the impact of population characteristics such as age, gender and curve severity on the use of OMs in AIS.”

Why the authors mention ‘with little exploration of the use of…….? It creates confusion in understanding the significance of this study. An appropriate statement is needed to avoid any confusion or misunderstanding to the researchers.

Results:

Results are systematically collected and are pooled in well-defined objectives.

The objectives are well-designed and the information about the barriers in collecting such data may help in to design a better approach to explore studies in a larger cohort with more improvisations.

Reviewer #2: Dear authors,

I appreciate you giving me the chance to read this excellent paper. I had a great time reading it. But I do have a few issues that need to be addressed.

Line 57 Page 2: The abbreviation should ideally be between brackets.

Line 114 Page 5: How did you preserve the autonomy and confidentiality of the participants?

How the information was kept safe and secure?

Line 119 Page 5: in line with (the COnsolidated criteria for REporting Qualitative studies) Please fix the capital, small letters issue

Line 119 Page 5: What kind of study design was applied? Could you perhaps add one or two sentences outlining the research design?

Line 147 Page 6: discuss the measures taken to maintain the study's rigor and trustworthiness.

Line 148 Page 6: Please add a diagram summarizing the data analysis steps

6. PLOS authors have the option to publish the peer review history of their article (what does this mean?). If published, this will include your full peer review and any attached files.

Reviewer #1: No

Reviewer #2: No

---

## [Author Response · Author response to Decision Letter 0]

19 Dec 2023

Reviewer #1: Overall comment:

Although the present study is having a very limited number of participants, this study has a significance for the HCPs that must be perceived as a valuable parameter for using Oms. The obtained results are of original research in this study and lay emphasis on the current practice of following and implementing HCPs. The study design and methodologies used here in are scientifically sound and are described in sufficient details. Furthermore, this study contributes to the aspect of encouraging all the participants (surgeons, nurses and physiotherapists etc.) to understand the importance of such data to overcome the barriers and improve the quality of life of individuals with AIS. Conclusions drawn upon by the authors conform in an appropriate fashion and are supported by relevant references. The ethics of experimentation and research integrity is satisfactorily applied in this study. The article adheres to appropriate reporting guidelines and community standards for data availability. Limited knowledge, and awareness to support the use PROMs by the healthcare individuals as well as the patients along with their facilitators, challenge such efforts and work as barriers that constrain PROMs use, while providing support and having quality PROMs facilitate its use to act as an enhancer to the use of performance measures among this population. Such works need to be carried out at a larger level and the present study can provide impetus to achieve this goal in near future.

Thank you for your thoughtful and constructive feedback, which acknowledges the significance of our study in highlighting the importance of using outcome measure by healthcare professionals among individuals with AIS.

Background:

The authors make two statements:

(lines 92-94) that “Nevertheless, variables that could improve or limit the usefulness of such measurements in individuals with AIS remain unknown.”

(lines 100-102) that “To date, no research has explored current practice of using OMs for individuals with AIS, or health care professionals’ (HCPs) perceived barriers and facilitators of all forms of OMs.”

These two statements stand diluted with their third statement:

(lines 98-100) that “Current research mostly focus on PROMs, with little exploration of the use of performance measures or the impact of population characteristics such as age, gender and curve severity on the use of OMs in AIS.”

Why the authors mention ‘with little exploration of the use of…….? It creates confusion in understanding the significance of this study. An appropriate statement is needed to avoid any confusion or misunderstanding to the researchers.

We have revised the sentence as follows 

The existing research predominantly emphasizes PROMs in individuals with AIS but lacks substantial exploration into the utilization of performance measures and the impact of demographic factors, such as age and curve severity, on the application of OMs in AIS. 

Page 4, Lines 97-100.

Results:

Results are systematically collected and are pooled in well-defined objectives.

The objectives are well-designed and the information about the barriers in collecting such data may help in to design a better approach to explore studies in a larger cohort with more improvisations

Thank you for positive comments.

Reviewer #2: 

Dear authors,

I appreciate you giving me the chance to read this excellent paper. I had a great time reading it. But I do have a few issues that need to be addressed.

We're delighted to hear that you enjoyed reading our paper, and we appreciate your valuable feedback.

Line 57 Page 2: The abbreviation should ideally be between brackets.

Brackets have been added. Page 2-line 59

Line 114 Page 5: How did you preserve the autonomy and confidentiality of the participants? How was the information kept safe and secure?

A paragraph has been added to the text detailing the process of assuring confidentiality of participants and storage of information. Page 6 lines 151 to 160.

Line 119 Page 5: in line with (the COnsolidated criteria for REporting Qualitative studies) Please fix the capital, small letters issue The abbreviation has now changed in the text. Page 5 line 122

Line 119 Page 5: What kind of study design was applied? Could you perhaps add one or two sentences outlining the research design? Qualitative research design has been used with descriptive phenomenology used as methodology embedded in pragmatic view of world. Sentences have been added to the text. Page 5 lines 121-125

Line 147 Page 6: discuss the measures taken to maintain the study's rigor and trustworthiness. In this research many steps were undertaken to ensure the study’s rigor and trustworthiness. A paragraph was added to manuscript page7-8, lines 184-192

Line 148 Page 6: Please add a diagram summarizing the data analysis steps A diagram showing data analysis has been added to the text. 

Page 7, line 173

---

## [Decision Letter · Decision Letter 1]

3 Jan 2024

Outcome Measures for Young People with Adolescent Idiopathic Scoliosis: A Qualitative Exploration of Healthcare Professionals' Perceptions and Practices

PONE-D-23-27580R1

Dear Dr. Alamrani,

We’re pleased to inform you that your manuscript has been judged scientifically suitable for publication and will be formally accepted for publication once it meets all outstanding technical requirements.

Kind regards,

Abera Mersha, MSc.

Academic Editor

PLOS ONE

Additional Editor Comments (optional):

Reviewers' comments:

Reviewer's Responses to Questions

**Comments to the Author**

1. If the authors have adequately addressed your comments raised in a previous round of review and you feel that this manuscript is now acceptable for publication, you may indicate that here to bypass the “Comments to the Author” section, enter your conflict of interest statement in the “Confidential to Editor” section, and submit your "Accept" recommendation.

Reviewer #2: All comments have been addressed

2. Is the manuscript technically sound, and do the data support the conclusions?

Reviewer #2: Yes

3. Has the statistical analysis been performed appropriately and rigorously? 

Reviewer #2: N/A

4. Have the authors made all data underlying the findings in their manuscript fully available?

Reviewer #2: Yes

5. Is the manuscript presented in an intelligible fashion and written in standard English?

Reviewer #2: Yes

6. Review Comments to the Author

Reviewer #2: Great work. I don’t have any further comments and wish the authors the very best of luck with their publication.

7. PLOS authors have the option to publish the peer review history of their article (what does this mean?). If published, this will include your full peer review and any attached files.

Reviewer #2: No

---

## [Editor Report · Acceptance letter]

18 Jan 2024

PONE-D-23-27580R1 

PLOS ONE

Dear Dr. Alamrani, 

I'm pleased to inform you that your manuscript has been deemed suitable for publication in PLOS ONE. Congratulations! Your manuscript is now being handed over to our production team.

Kind regards, 

on behalf of

Mr. Abera Mersha 

Academic Editor

PLOS ONE